# An ankyrin-repeat and WRKY-domain-containing immune receptor confers stripe rust resistance in wheat

Huan Wang [1,2], Shenghao Zou[1], Yiwen Li [2], Fanyun Lin [2] & Dingzhong Tang[1✉]

Perception of pathogenic effectors in plants often relies on nucleotide-binding domain (NBS) and leucine-rich-repeat-containing (NLR) proteins. Some NLRs contain additional domains that function as integrated decoys for pathogen effector targets and activation of immune signalling. Wheat stripe rust is one of the most devastating diseases of crop plants. Here, we report the cloning of *YrU1*, a stripe rust resistance gene from the diploid wheat *Triticum urartu*, the progenitor of the A genome of hexaploid wheat. *YrU1* encodes a coiled-coil-NBS-leucine-rich repeat protein with N-terminal ankyrin-repeat and C-terminal WRKY domains, representing a unique NLR structure in plants. Database searches identify similar architecture only in wheat relatives. Transient expression of YrU1 in *Nicotiana benthamiana* does not induce cell death in the absence of pathogens. The ankyrin-repeat and coiled-coil domains of YrU1 self-associate, suggesting that homodimerisation is critical for YrU1 function. The identification and cloning of this disease resistance gene sheds light on NLR protein function and may facilitate breeding to control the devastating wheat stripe rust disease.

[1] State Key Laboratory of Ecological Control of Fujian-Taiwan Crop Pests, Key Laboratory of Ministry of Education for Genetics, Breeding and Multiple Utilization of Crops, Plant Immunity Center, Fujian Agriculture and Forestry University, Fuzhou 350002, China. [2] State Key Laboratory of Plant Cell and Chromosome Engineering, Institute of Genetics and Development Biology, Chinese Academy of Sciences, Beijing 100101, China. ✉email: dztang@genetics.ac.cn

  1

Wheat stripe rust, caused by *Puccinia striiformis* f. sp. *tritici* (*Pst*), is one of the most devastating diseases of crops, causing major production losses in hexaploid wheat (*Triticum aestivum* L.) around the world[1]. Stripe rust has become the most damaging of all the crop rusts due to its expanding range and the resulting increased production losses[2]. To date, over 80 stripe rust resistance genes have been identified and mapped in wheat[3], but only a few, including *Yr5, Yr7, YrSP, Yr15, Yr18/Lr34, Yr36, Yr46* and *YrAS2388*, have been cloned[4–9]. Among these, *Yr18/Lr34* encodes a putative ATP-binding cassette (ABC) transporter; *Yr36* encodes a protein with a kinase domain and a putative lipid-binding domain; *Yr46* encodes a hexose transporter; and *Yr15* encodes a protein with predicted kinase and pseudokinase domains. These genes encode different protein families and confer a relatively broad spectrum of resistance. In contrast, *YrAS2388* encodes a nucleotide-binding site (NBS) and leucine-rich repeat (LRR) proteins (NLRs), and *Yr5, Yr7* and *YrSP* encode NLR with additional BED domain; all of these genes are race specific, being effective against only a subset of *Pst* isolates at the seedling or all stages[4–9]. The robustness of stripe rust resistance genes has been challenged by coevolution between wheat and stripe rust pathogens in which the pathogens have evolved the capacity to overcome the resistance conveyed by these genes[2,10]. For instance, *Yr7*, which confers resistance to many stripe rust isolates, has been overcome by newer isolates[11,12].

Plants use intracellular NLRs to detect pathogens by direct or indirect recognition of pathogen effectors[13]. Most NLRs contain three domains: an N-terminal variable domain, NBS domain and C-terminal LRR[13,14]. Several also have additional functional domains, which are hypothesised to act as integrated decoys. For instance, the *Arabidopsis thaliana* (Arabidopsis) resistance protein RRS1 has a C-terminal WRKY domain; the rice (*Oryza sativa*) resistance protein RGA5 has a C-terminal RATX1 domain; the wheat stripe rust resistance proteins Yr5, Yr7 and YrSP have an N-terminal BED domain; and the tomato resistance proteins Sw-5b and Prf have an extra N-terminal domain[7,15–18]. Among these, the RATX1 domain of RGA5 directly binds to a *Magnaporthe oryzae* effector, and activates RGA4-mediated resistance and cell death upon recognition[17,19]. The WRKY domain of RRS1 functions as a decoy to bind the pathogen effector and triggers immune signalling mediated by its neighbouring NLR RPS4[20,21]. The additional domains in other NLRs are not well characterised.

The diploid wheat species *T. urartu* is the progenitor species of the A genome of common wheat and a potential source for stripe rust resistance genes[22,23]. The high-quality genome sequence of *T. urartu* has been published and can be used for map-based cloning of genes[24,25]. In this study, we map-based cloned a stripe rust resistance gene from the resistant *T. urartu* accession PI428309. The gene encodes a CC-NBS-LRR protein with an N-terminal ankyrin-repeat (ANK) domain and a C-terminal WRKY domain. The ANK-NLR-WRKY domain is rare, and this type of protein/gene structure is found only in the genomes of wheat and its relatives. The identification and cloning of this type of stripe rust resistance gene will not only help to uncover new mechanism of disease resistance but also facilitate breeding for disease resistance in wheat.

## Results

***Triticum urartu* PI428309 is resistant to stripe rust.** To identify stripe rust resistance genes in *T. urartu*, we infected a large number of *T. urartu* accessions at the seedling stage with the *Puccinia striiformis* f. sp. *tritici* (*Pst*) isolate CYR33 and observed the disease severity 14 days later. G1812, the accession used for the reference genome sequence[24], was highly susceptible to *Pst*

CYR33, displaying a large number of visible uredia (spore-bearing pustules characteristic of rust disease) and lacked visible cell death (which is indicative of a resistance response) on the leaves at 14 days post inoculation (dpi; Fig. 1a, Supplementary Fig. 1). However, several *T. urartu* accessions showed resistance to *Pst* CYR33, with very few or no uredia and prominent cell death 14 dpi (Supplementary Fig. 1).

We selected for further characterisation the accession PI428309, which showed resistance to five stripe rust races, CYR17, CYR31, CYR32, CYR33, and V26 (Fig. 1a, Supplementary Fig. 2), and was recently used to clone a powdery mildew resistance gene, *Pm60*[26]. At 2 dpi, after wheat germ agglutinin (WGA) staining, fluorescence micrographs of the fungal structures showed that substomatal vesicles (SSV), primary infection hyphae (PH) and haustoria mother cells (HMCs) were formed in both PI428309 and the susceptible accession G1812, and haustoria were already found in G1812, but not in PI428309 (Fig. 1b). At 7 dpi, fewer fungal colonies and pathogen feeding structures were observed in PI428309 than in G1812 (Fig. 1c). We examined the transcript levels of several pathogenesis-related genes and found that, consistent with the resistant phenotype in PI428309, the pathogenesis-related genes *PR1, PR2, PR3,* and *PR5* were induced to much higher levels in PI428309 than in G1812 (Supplementary Fig. 3), indicating that resistance responses were strongly activated in PI428309.

To study the genetic basis of stripe rust resistance in PI428309, we crossed PI428309 and G1812 and evaluated the disease resistance response of the $F_1$ plants when infected with *Pst* CYR33. The $F_1$ plants were resistant to CYR33, producing limited uredia and showing prominent cell death on the leaves at 14 dpi (Fig. 1); their infection types were similar but slightly higher than those of the PI428309 plants. Next, we inoculated the $F_2$ plants with *Pst* CYR33 and phenotyped individual plants for stripe rust resistance. The segregation ratio of disease resistance in the $F_2$ generation showed that the resistance to *Pst* CYR33 in PI428309 was controlled by one major dominant resistance locus, which we tentatively named *Yr* (Supplementary Table 1).

**Genetic and physical maps of the resistance gene.** To map the *Yr* stripe rust resistance gene in PI428309, we screened 259 simple sequence repeat (SSR) markers on the common wheat chromosomes (1A to 7A) to analyse the polymorphisms between PI428309 and G1812. Eighty-two polymorphic markers were used to screen the susceptible plants of the $F_2$ population derived from the cross between PI428309 and G1812. We discovered that marker *Xgpw7007-5A* (Fig. 2a), in the wheat deletion-line bin 5AL10-0.57-0.78 of the *T. aestivum* cultivar Chinese Spring, was linked with the stripe rust resistance locus.

To identify more molecular markers tightly linked to the stripe rust resistance locus, we used the wheat expressed sequence tags (ESTs) in the deletion bins 5AL12-0.35-0.57 and 5AL10-0.57-0.78 to search the genome sequences of G1812 using BLASTN. We then used the scaffolds of G1812 in the deletion-line bins to develop SSR markers. Through this method, we developed several markers closely linked with the stripe rust resistance locus *Yr*, including SCF12 (from scaffold 68558), SCF13 (from scaffold 21752), SCF19 (from scaffold 9979) and SCF20 (from scaffold 49434) (Fig. 2a, Supplementary Table 2). To further identify new markers, we used the *Brachypodium distachyon* genes collinear to *Yr* to search the genome sequences of G1812 using BLASTN, and then develop additional SSR and cleaved amplified polymorphic sequence (CAPS) markers with those sequences (Supplementary Table 3). Using 3,034 susceptible plants, identified among a total of 11,906 $F_2$ plants from the cross between PI428309 and G1812, we ultimately anchored the *Yr* resistance locus to a region of

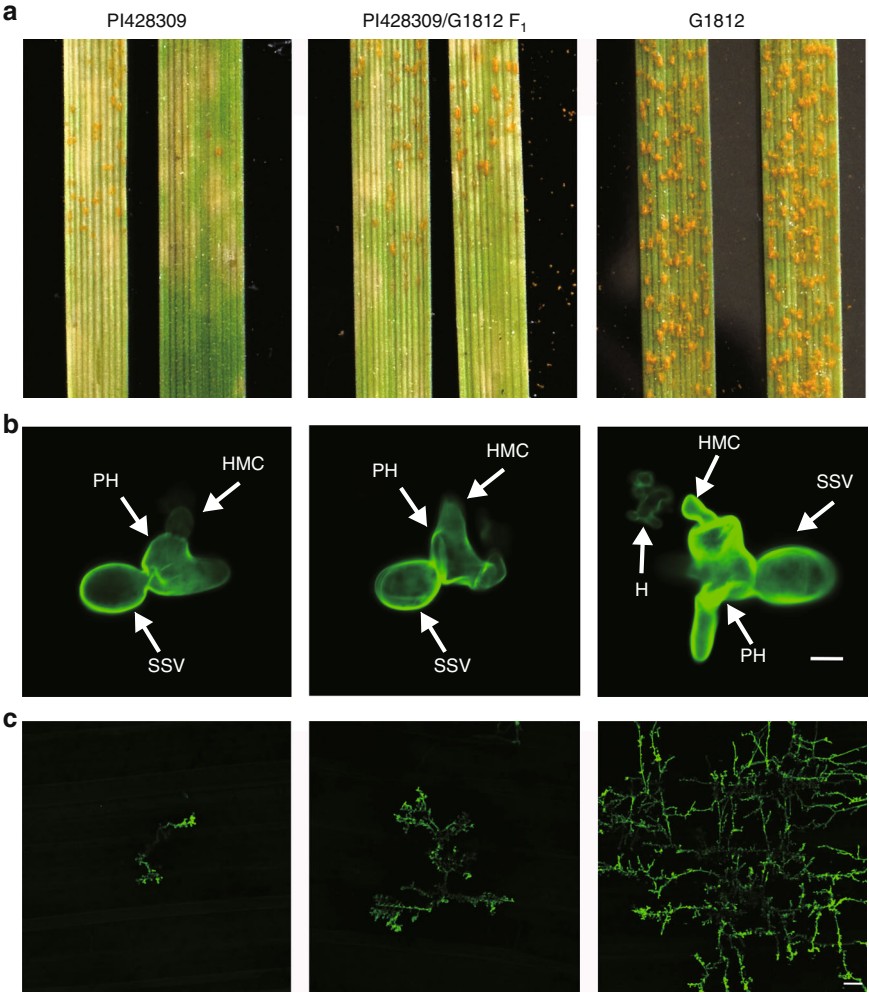

**Fig. 1 *T. urartu* PI428309 showed resistance to stripe rust pathogen *Puccinia striiformis* f. sp. *tritici* (*Pst*) CYR33. a** Ten-day-old plants of accessions PI428309 and G1812 and F₁ progeny of a cross between PI428309 and G1812 were infected with *Pst* CYR33. Plant leaves were detached and photographed at 14 dpi. PI428309 (infection type (IT) 1) and F₁ (IT 2) showed resistance to *Pst* CYR33, while G1812 (IT 4) showed susceptibility to *Pst* CYR33. **b, c** Fluorescence micrographs of the fungal structures of *Pst* CYR33 at 2 and 7 dpi in PI428309, G1812 and F₁ plants. The fungal structures were stained with wheat germ agglutinin (WGA). SSV, substomatal vesicle; PH, primary infection hyphae; HMC, haustoria mother cells; H, haustoria. Bars, 10 μm (**b**), 100 μm (**c**).

65 kb in the G1812 chromosome 5AL (Fig. 2c). This region contains three candidate genes, *TuG1812G0500003718*, *TuG1812G0500003719* and *TuG1812G0500003720*, in G1812.

**Cloning of the stripe rust resistance gene in PI428309.** *TuG1812G0500003718*, *TuG1812G0500003719* and *TuG1812G0500003720* all encode NLRs. We obtained two candidate genes in PI428309 allelic to *TuG1812G0500003718* and *TuG1812G0500003719*, which we designated *YrU1* and *CG2* based on subsequent analysis (Supplementary Data 1). There is no allelic gene of *TuG1812G0500003720* in PI428309 because of sequence differences between PI428309 and G1812. We analysed the sequences of *YrU1* and *CG2* in PI428309 and G1812 and found that *YrU1* has ~1 Kb of additional sequence in PI428309 as compared with G1812 (Supplementary Fig. 4). There were no significant sequence differences in *CG2* between PI428309 and G1812 except for a few single-nucleotide polymorphisms (SNPs, Supplementary Fig. 5). The additional ~1 Kb in PI428309 *YrU1* encode a WRKY domain that is absent in G1812 (Supplementary Figs. 4 and 6). We designed a marker based on this 1 Kb sequence to screen for its presence in different *T. urartu* accessions and found that this sequence was present only in resistant

accessions, but not in any susceptible accessions we examined (Supplementary Fig. 4). Among 157 *T. urartu* accessions investigated, those containing the additional sequence detected by this marker were all resistant to stripe rust *Pst* CYR33 (Supplementary Table 4).

We also examined the transcript levels of *YrU1* and *CG2* in PI428309 and G1812 before and after inoculation with *Pst* CYR33 by quantitative reverse transcription PCR (qRT-PCR). The expression of *YrU1* was only slightly affected by infection (Supplementary Fig. 7), but the expression of *CG2* was not detected either before or after inoculation in either G1812 or PI428309. Based on these results, we selected *YrU1* for further investigation as the candidate gene.

**YrU1 knockdown suppresses PI428309 resistance to *Pst* CYR33.** To determine whether *YrU1* is responsible for resistance to stripe rust *Pst* CYR33 in PI428309, we transiently silenced *YrU1* in *T. urartu* PI428309 using the *Barley stripe mosaic virus* (BSMV)-induced gene silencing system. This silencing impaired the plants' resistance to stripe rust *Pst* CYR33. At 14 dpi, the leaf surfaces of BSMV:*YrU1*-treated plants displayed numerous visible uredia, in contrast with the BSMV:*GFP* control plants, which had

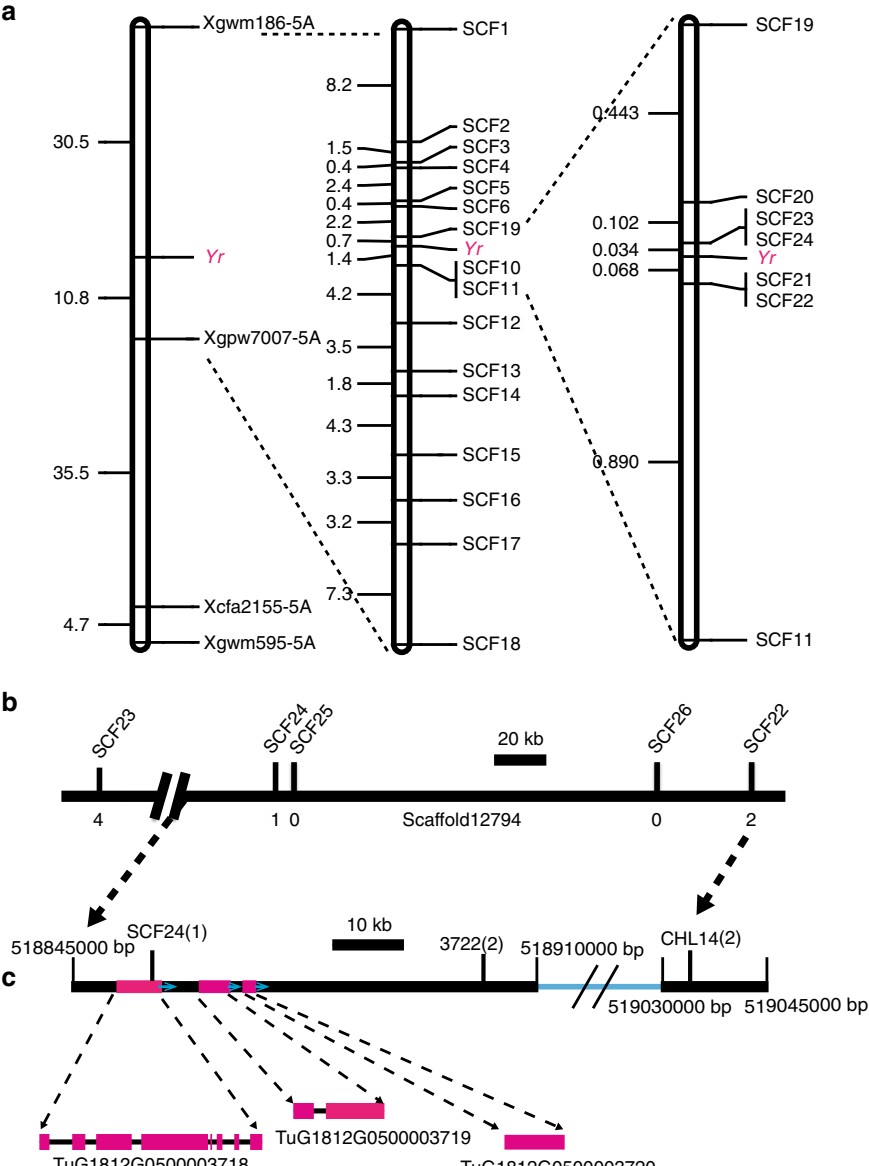

**Fig. 2 Genetic and physical maps of the locus conferring resistance to *Pst* CYR33. a** Genetic maps of the locus conferring resistance to *Pst* CYR33 constructed using F$_2$ populations derived from the cross between PI428309 and G1812. **b** Physical maps of *Yr* locus on scaffold12794. The number of recombinants are indicated below the scaffold12794. **c** Physical maps of *Yr* locus on 518845000 to 519045000 bp of chromosome arm 5AL. The blue line corresponds to 518910000 to 519030000 bp region on the G1812 chromosome arm 5AL. Three genes (*TuG1812G0500003718, TuG1812G0500003719* and *TuG1812G0500003720*) found at the stripe rust resistance (*Yr*) mapping locus of G1812.

few uredia (Fig. 3). This result indicated that *YrU1* is responsible for resistance to *Pst* CYR33 in PI428309.

**Validation of *YrU1* gene by transgenic complementation.** To further test whether *YrU1* is responsible for conferring resistance to stripe rust *Pst* CYR33, we cloned a genomic fragment of *YrU1* from *T. urartu* PI428309, comprising 2714 bp upstream of the start codon, the coding region and 2086 bp downstream of the stop codon, into the binary vector pCAMBIA-1300. We then introduced the construct into the stripe-rust-susceptible wheat cultivar Bobwhite. We obtained six independent transgene-positive plants, and PCR and qRT-PCR analysis confirmed the presence and expression of the transgene in the T$_0$, T$_1$ and T$_2$ progeny of three representative transgenic lines (Fig. 4). The infection assays in the T$_1$ and T$_2$ plants demonstrated that the transgenic plants expressing *YrU1* supported only limited uredia

and showed prominent cell death at 14 dpi, in contrast to Bobwhite and to T$_1$ plants that did not express *YrU1*, which produced abundant uredia and lacked prominent cell death (Fig. 4b). T$_2$ plants expressing *YrU1* also showed resistance to stripe rust races CYR17, CYR32 and V26, and produced fewer uredia than Bobwhite in all cases (Supplementary Fig. 8). Taken together, these results indicate that *YrU1* confers resistance to *Pst* races CYR33, CYR17, CYR32 and V26 in wheat.

***YrU1* encodes an integrated ANK-NLR-WRKY immune receptor.** The *YrU1* gene encodes a CC-NBS-LRR protein, with an additional ANK domain at the N terminus and a WRKY domain at the C terminus. Unlike previously cloned integrated NLR immune receptors (such as RRS1, RGA5, Yr7, Yr5, YrSP and Sw-5b[7,15,17,21]), YrU1 has integrated domains at both its C and N termini. The domain structure of the YrU1 protein is very

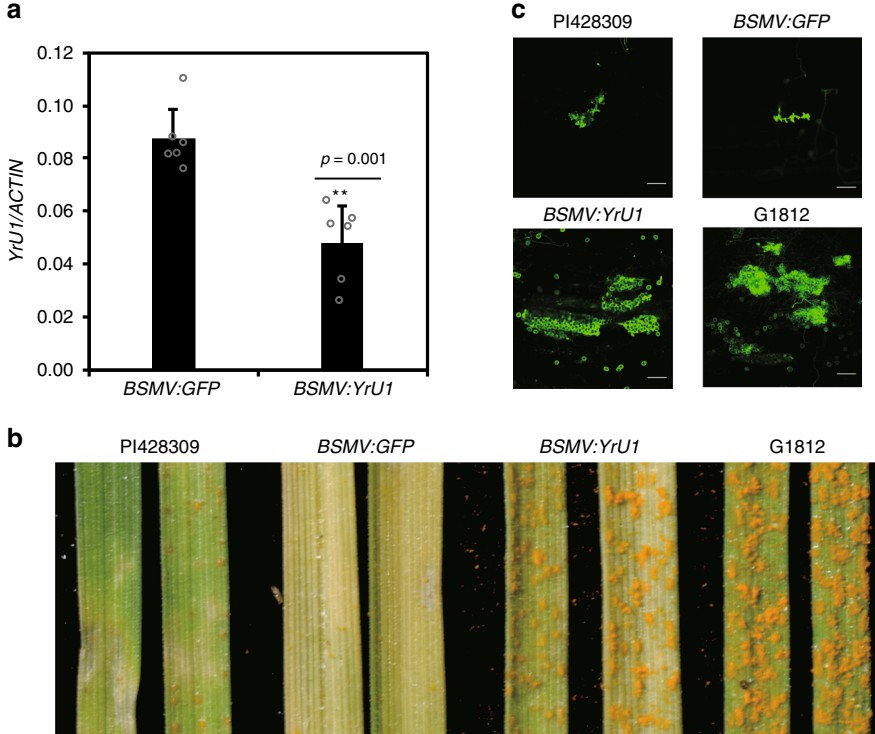

**Fig. 3 Barley stripe mosaic virus (BSMV)-induced gene silencing of _YrU1_ suppresses the resistance of PI428309 to _Puccinia striiformis_ f. sp. _tritici_ (_Pst_) CYR33. a** Relative transcript levels of _YrU1_ in _BSMV:GFP_ (control) and _BSMV:YrU1_ plants were examined by qRT-PCR. Results represent the means ± SD from six independent biological samples. Two asterisks indicate statistically significant difference (_P_ < 0.01, One-Way ANOVA). **b** PI428309, _BSMV: GFP_, _BSMV:YrU1_ and G1812 plants were inoculated with _Pst_ CYR33 and photographed at 14 dpi. **c** Fluorescence micrographs of the fungal structures of _Pst_ CYR33 at 14 dpi in PI428309, _BSMV:GFP_, _BSMV:YrU1_ and G1812 plants. The fungal structures were stained with wheat germ agglutinin (WGA). Bars, 100 μm. This experiment was repeated at least three times with similar results.

unusual: we identified only three ANK-NLR-WRKY genes across the genomes of all species from the public reference proteome database UniProt. Those three ANK-NLR-WRKY genes are from _T. urartu_ (_YrU1_), _T. aestivum_ and emmer wheat (_Triticum dicoccoides_). We also performed a BLASTP analysis with the YrU1 protein in _T. durum_ Svevo whole genome and did not find any ANK-NLR-WRKY protein, indicating that _YrU1_-type genes exist only in _Triticum_ species (Fig. 5b, Supplementary Table 5). Sequence alignment indicated that those three proteins have >90% sequence identity and the genes are likely to be homologous. There are only two amino acid differences between the two ANK-NLR-WRKY proteins from the wheat cultivar Chinese Spring and the wild emmer accession "Zavitan" (Fig. 5a). The sequences of YrU1 are more divergent from the Chinese Spring and Zavitan proteins than those two are from each other, and the differences between YrU1 and the other two ANK-NLR-WRKY proteins exist mainly in the WRKY domains (Fig. 5a). Although Chinese Spring contains an ANK-NLR-WRKY gene that is highly similar to _YrU1_, the cultivar is susceptible to the _Pst_ CYR33 (Supplementary Fig. 9), indicating that the YrU1 variant in Chinese Spring is not functional in regard to resistance to _Pst_ CYR33.

Beside those three ANK-NLR-WRKY genes, we also identified one ANK-NLR gene (the allelic gene of _YrU1_ in G1812) and 17 NLR-WRKY genes across the wheat genomes and the genomes of the related grass species _Aegilops tauschii_, _Triticum urartu_, _Hordeum vulgare_ (barley), _Brachypodium distachyon_, _Oryza sativa_ (rice) and _Zea mays_ (corn) (Fig. 5b, Supplementary Table 5). Sequence alignment of the WRKY domains from the 17 NLR-WRKY proteins indicated that the WRKY heptad domains are relatively conserved (black line in Supplementary Fig. 10). The

sequence of this domain of YrU1 is consistent with that of Arabidopsis RRS1. The WRKY heptad domain is detected in acetylated peptides in RRS1, and the K residues of this domain are necessary for recognition of PopP2 and AvrRps4[20,21]. Those results suggest that YrU1 may recognise and bind stripe rust effector through the WRKY domain.

We identified 54 additional ANK-domain-containing proteins and 73 WRKY-domain-containing proteins across the _T. urartu_ genome (Supplementary Tables 6 and 7). We constructed a phylogenetic tree based on 56 ANK-containing proteins from _T. urartu_ and two ANK-NLR-WRKY proteins from _T. aestivum_ and _T. dicoccoides_, respectively, and a second phylogenetic tree based on 94 WRKY-containing proteins, including 76 from _T. urartu_, 17 NLR-WRKYs from grasses and RRS1 from Arabidopsis. The dendrogram based on 58 ANK-containing proteins showed that ANK domains are diverse, but are split from ANK-NLR proteins and non-NLR proteins (Fig. 5c). Those results suggest that the integration of ANK domains was most probably derived from one integration event. Similarly, the dendrogram based on 94 WRKY-containing proteins (except for _BGIOSGA022298-PA_ from rice) show that WRKY domains are diverse, but are split from NLR-WRKY proteins and non-NLR proteins (Fig. 5d).

**YrU1 overexpression does not induce cell death.** To further characterize the function of YrU1, we transiently expressed YrU1 with an N-terminal GFP tag or C-terminal HA tag under the control of the _35S_ promoter in _Nicotiana benthamiana_. Overexpression of full-length YrU1 in _N. benthamiana_ leaves did not induce cell death. In contrast, the positive control, Pm60 (a protein that conveys resistance to powdery mildew), did induce cell death in _N. benthamiana_ leaves (Supplementary Fig. 11).

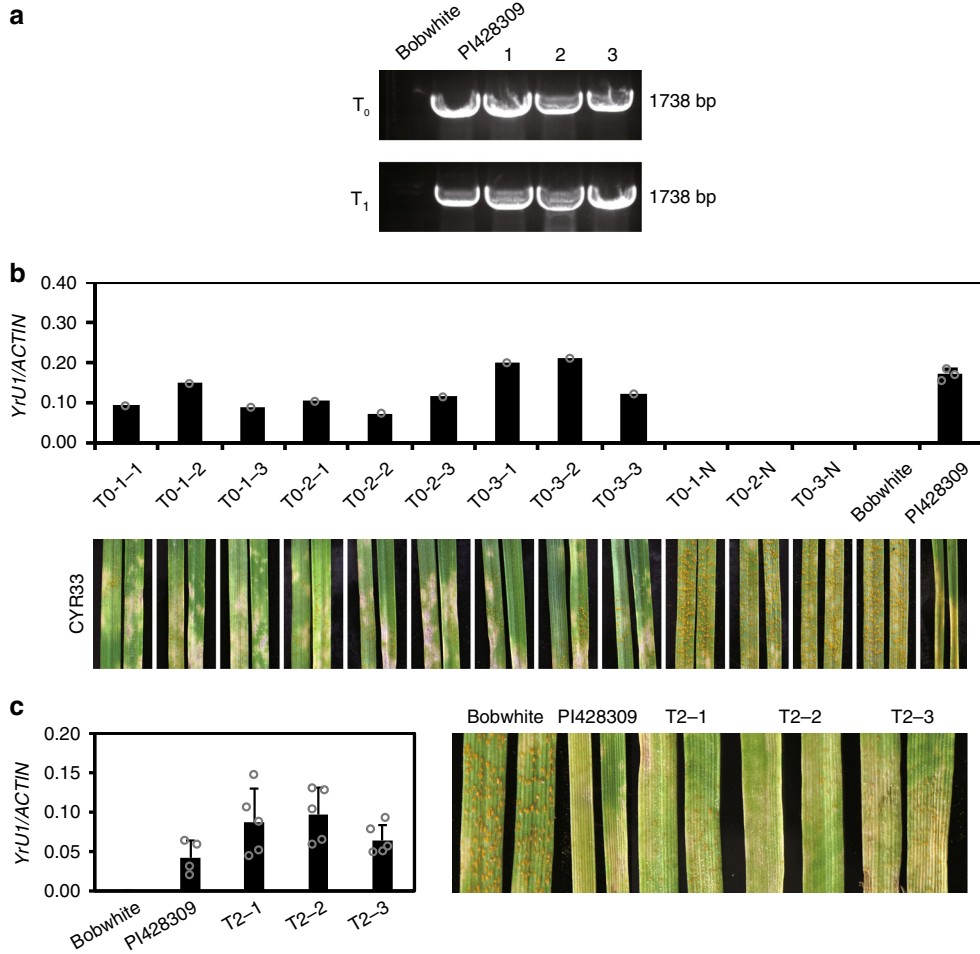

**Fig. 4 Validation of *YrU1* by transgenic complementation. a** Detection of *YrU1* gene in susceptible bread wheat cv. Bobwhite and PI428309 and in $T_0$ and $T_1$ transgenic plants from three independent lines (Bobwhite background) by PCR. **b** Transcript levels of *YrU1* as determined by qRT-PCR in $T_1$ transgenic plants from three independent lines (Bobwhite background) and their resistance to *Pst* CYR33 were evaluated at the seedling stage. T0-1-1, T0-1-2, T0-1-3 and T0-1-N were from the T0-1 line; T0-2-1, T0-2-2, T0-2-3 and T0-2-N were from the T0-2 line; T0-3-1, T0-3-2, T0-3-3 and T0-3-N were from the T0-3 line; and T0-1-N, T0-2-N, T0-3-N and Bobwhite were used as negative controls. PI428309 was used as the positive control. **c** Transcript levels of *YrU1* and resistance to *Pst* CYR33 in $T_2$ plants from three independent lines. Ten-day-old plants were infected with *Pst* CYR33. The leaves were detached and photographed at 14 dpi. Error bars represent the SD from at least three independent biological samples. All experiments were repeated at least three times with similar results.

Next, we overexpressed different domains of YrU1, including the ANK, CC, NB-ARC, LRR and WRKY domains, in *N. benthamiana* leaves, and found that none of these domains induced cell death (Supplementary Fig. 11). Taken together, those results suggest that YrU1 alone does not induce cell death in *N. benthamiana* leaves, and cell death may require the presence of pathogen effector or other proteins, similar to what has been reported previously for other NLR proteins with integrated domains, such as RRS1 and RGA5[19,20].

**YrU1 ANK and CC domains self-associate**. Previous studies reported that self-association of the N terminus of NLR proteins plays a crucial role in triggering downstream immune signals[27–29]. YrU1 contains an N-terminal ANK domain, a type of domain that typically functions in mediating specific protein–protein interactions[30]. This implied that the N-terminal domain of YrU1 might self-associate, as reported previously for other NLR proteins, such as MLA10, Sr33, Sr50, RGA5 and L6[19,27,29]. To validate this hypothesis, we examined whether N-terminal domains of YrU1 self-associate using yeast two-hybrid, luciferase complementation imaging and co-immunoprecipitation assays (Fig. 6). The results

showed that the ANK domain and CC domain of YrU1 can indeed self-associate in vivo and in planta. These results suggest that YrU1 can form N-terminally linked homodimers, a process that might play a role in transducing immune signals.

## Discussion

The wild relatives of wheat are good sources of resistance genes, and many such genes have been cloned, such as *Sr33*, *Pm21* and *Yr36*[4,31,32]. Here, we cloned a stripe rust resistance gene, *YrU1*, from *T. urartu* accession PI428309 and determined that it conveys resistance to several races of stripe rust pathogen in common wheat (Supplementary Fig. 8). Therefore, the *YrU1* gene is a potentially valuable tool for use in breeding wheat strains for cultivation.

YrU1 is a NLR protein that contains an N-terminal ANK and a C-terminal WRKY domain. ANK-NLR-WRKY proteins are rare: we identified only three ANK-NLR-WRKY proteins from the public reference proteome databases UniProt and Ensemble. All three are found in *Triticum* species, and their sequences are highly conserved. The common wheat cultivar Chinese Spring, which contains an ANK-NLR-WRKY protein, is susceptible to

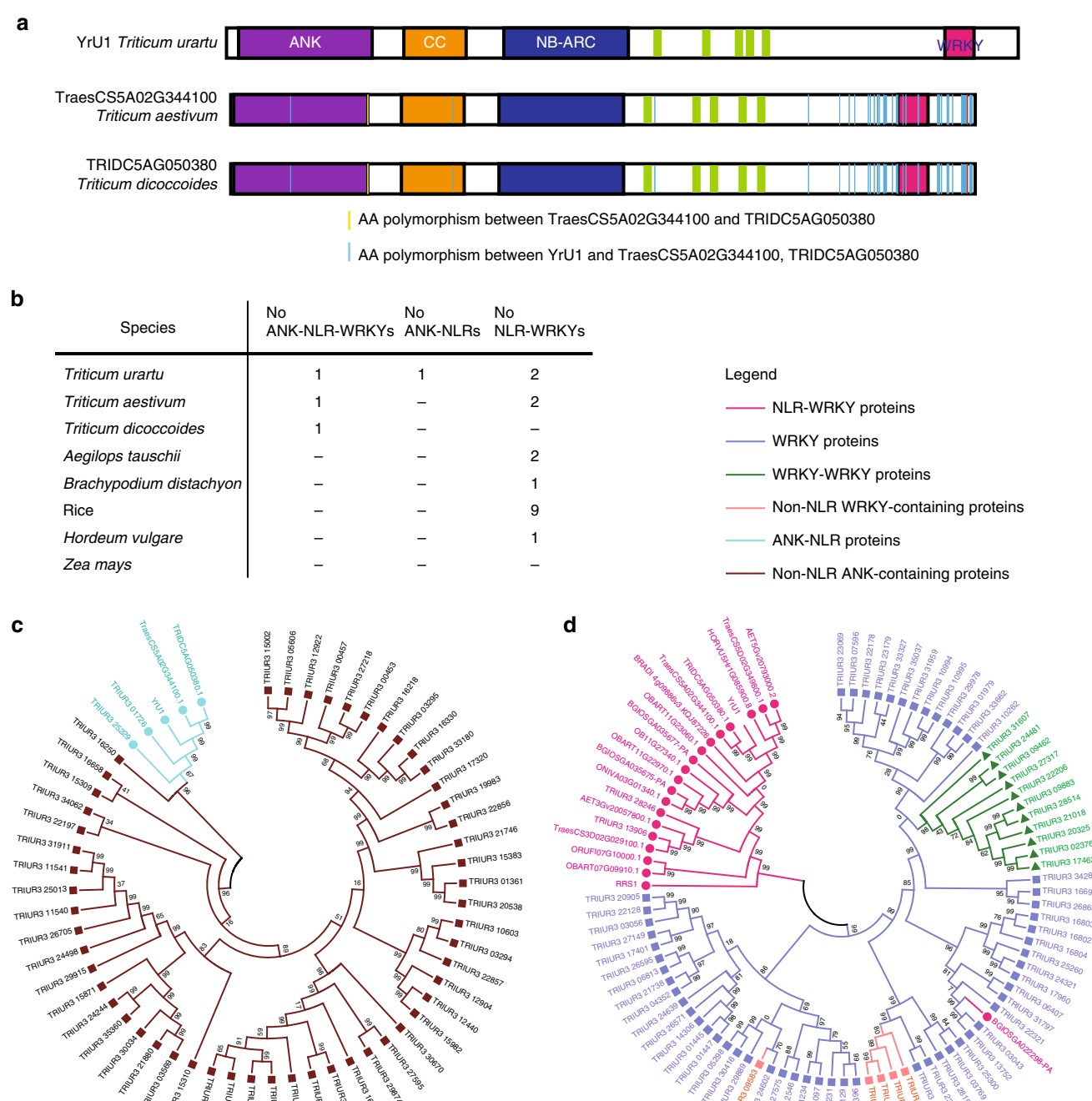

**Fig. 5 YrU1 encodes an integrated ANK-NLR-WRKY immune receptor. a** Schematic representation of YrU1 protein domain organisation. *TraesCS5A02G344100* and *TRIDC5AG050380*, from *T. aestivum* and *T. dicoccoides*, respectively, also encode ANK-NLR-WRKY proteins. ANK domains are highlighted in purple, CC domains in yellow, NB-ARC domains in blue, LRR motifs in green and WRKY domains in red. Amino acid polymorphisms between YrU1 and the other two ANK-NLR-WRKY proteins (TraesCS5A02G344100 and TRIDC5AG050380) are highlighted with light blue vertical bars, and amino acid polymorphisms of TraesCS5A02G344100 and TRIDC5AG050380 with red vertical bars. Matching colours on the protein structure indicate 100% sequence conservation. **b** Numbers of ANK-NLR-WRKY, ANK-NLR and NLR-WRKY genes in different grass genomes (see Supplementary Table 5). **c** Phylogenetic analysis of 58 ANK-containing proteins including the 56 ANK-containing proteins from *T. urartu* and two ANK-NLR-WRKY proteins from *T. aestivum* and *T. dicoccoides*, respectively. ANK-NLRs are in light blue and non-NLR ANK-containing proteins are in brown. **d** Phylogenetic tree analysis with 94 WRKY domains including the 76 WRKY-containing proteins from *T. urartu*, 17 NLR-WRKYs from grass genomes and RRS1 from Arabidopsis. NLR-WRKYs and non-NLR-WRKY-containing proteins are highlighted in red and orange, respectively. WRKY and WRKY-WRKY proteins are in purple and green, respectively.

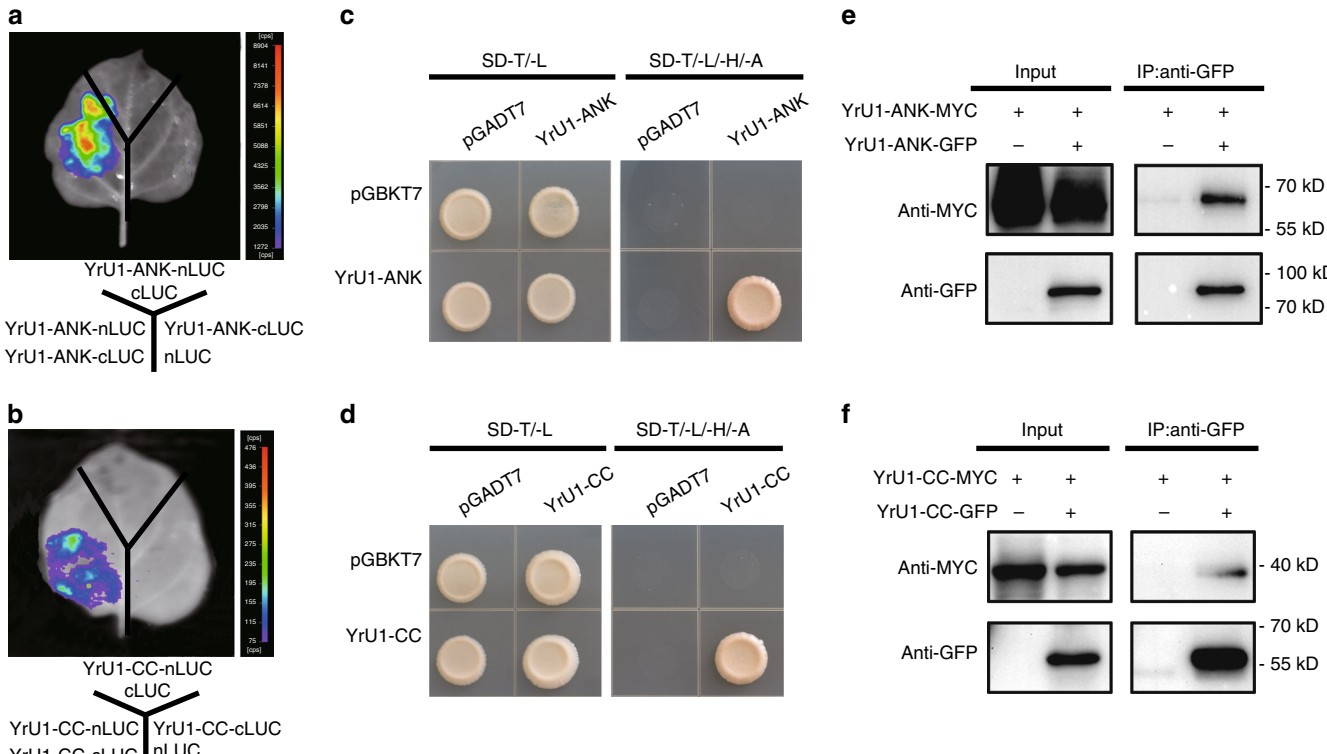

**Fig. 6 Self-association of the ANK and CC domains of YrU1. a, b** Self-association of ANK (**a**) and CC (**b**) domains of YrU1 in firefly luciferase complementation imaging (LUC) assay. **c, d** Self-association of ANK (**c**) and CC (**d**) domains of YrU1 in yeast two-hybrid assays. The coding sequences of YrU1-ANK and YrU1-CC were fused to the Gal4 transactivation domain (AD) or the Gal4 DNA-binding domain (BD). Eight pairs of constructs were co-transformed into AH109. A 10-µL suspension (OD$_{600}$ = 0.5) of each co-transformant was dropped onto synthetic dropout (SD) medium lacking Leu and Trp and SD medium lacking Ade, His, Leu and Trp. Photographs were taken after 3 d of incubation. **e, f** The ANK (**e**) and CC (**f**) domains of YrU1 can self-associate in planta. Shown are immunoblots with anti-GFP and anti-MYC antibodies of total proteins extracted at 48 h post inoculation (hpi) from *N. benthamiana* leaves transiently expressing YrU1-ANK-MYC, YrU1-ANK-GFP, YrU1-CC-MYC or YrU1-CC-GFP and immunoprecipitated with anti-GFP. All experiments were repeated at least two times with similar results.

stripe rust (*Pst*) race CYR33, indicating that the allelic *YrU1* in Chinese Spring is nonfunctional against *Pst* CYR33. The high level of identity of the three *Triticum* ANK-NLR-WRKY proteins suggests that they derive from a common evolutionary origin and that the *YrU1* variant in Chinese Spring lost the resistance to stripe rust during the evolution of the rust strains. This type of NLR protein is unique to wheat-related families and may reflect the existence of a specific NLR-mediated disease resistance mechanism in wheat relatives.

Several NLR resistance proteins reported previously have additional domains, such as the *Arabidopsis* resistance protein RRS1, with a C-terminal WRKY domain; rice blast resistance protein RGA5, with a C-terminal RATX1 domain; tomato resistance protein Sw-5b has an extra N-terminal domain; and wheat stripe rust resistance proteins Yr5, Yr7 and YrSP, with N-terminal BED domains[7,15,18,19]. YrU1 is the first NLR protein reported with extra domains at both the N and C termini. YrU1's N-terminal ankyrin repeats, a 33-amino-acid sequence motif, are generally involved in protein–protein interactions[33]. The best-characterized ANK protein from plants is Arabidopsis NPR1, which is a master regulator of the salicylic acid (SA) signalling pathway and functions as a transcriptional coactivator to regulate plant defence response[34,35]. The SUMO-interaction motif (SIM3) located in the ANK domain of NPR1 is modified by SUMO3 to regulate the defence response[36]. *Pseudomonas syringae* type III effector AvrPtoB targets NPR1 and subverts plant innate immunity by repressing NPR1-dependent SA signalling[37]. ACD6 and BDA1 are also ANK proteins and function as positive

regulators of SA signalling in defence responses[38–40]. Although YrU1 contains an ANK domain, it does not show high similarity to NPR1, ACD6 or BDA1. YrU1 is an NLR protein, whose ANK domain may interact with other NLR proteins or act as an integrated decoy domain to interact with the effector of the pathogen, thereby inducing plant immunity.

The C-terminal WRKY domain of YrU1 is a putative transcriptional domain. The *Arabidopsis* resistance protein RRS1-R is a TIR-NBS-LRR protein with a C-terminal WRKY domain that acts as an integrated decoy to detect and bind the bacterial effectors PopP2 and AvrRps4 to trigger the plant immunity[20,21]. Thus, the YrU1 protein, which has a WRKY domain similar to that of RRS1-R, may recognise and bind stripe rust effector through that domain, thereby activating immune responses.

Here, we showed that the ANK and CC domains of YrU1 self-associate. However, neither the individual YrU1 domains nor full-length YrU1 induced cell death in *N. benthamiana* leaves. NLR oligomerization via N-terminal domains, which are commonly coiled-coil or TIR domains in plants, is thought to initiate disease resistance signals by recruiting or activating downstream signals[13,41]. Indeed, previous studies of several NLR proteins, such as MLA10, Sr33, Sr35, N, L6, RRS1-R/RPS4 and RGA4/RGA5, indicate that their N-terminal domains can self-associate[19,20,27,29,42]. MLA10, Sr33, Sr35 and L6 can induce an effector-independent cell death response in planta[27,29]. Recent structural work has also shown that oligomerisation is critical for NLR activation[43,44]. However, RRS1-R or RGA5 triggering of cell death requires their partners RPS4 or RGA4 and pathogen

effectors[19,20]. YrU1, like RPS4 or RGA4, may require other NLR proteins and pathogen to trigger the defence responses.

It is possible that YrU1, like previously identified NLR proteins (such as RRS1 and RGA5), initiates the resistance response by recognising and binding a stripe rust effector through integrated decoy domains. However, YrU1 differs from these other NLR proteins in having integrated domains at both its C and N termini, and YrU1-type proteins exists only in wheat and its relatives. These two domains may thus play different roles in activating resistance responses than other NLR proteins do, and YrU1-mediated disease resistance may represent a mechanism present only in wheat. Activation of YrU1 may represent a unique molecular mechanism in NLR-mediated plant immunity. Based on current information, we present a working model of how YrU1 functions (Supplementary Fig. 12). In this model, YrU1 is in the rest state in the absence of pathogen. When stripe rust pathogen is present, the effector binds to the WRKY domain, resulting in conformational changes and oligomerisation of YrU1, which leads to activation of disease resistance. The ANK domain may recruit additional components to activate downstream signalling, or it may also act as a decoy for effector binding. It would be very interesting to determine the roles of the integrated ANK and WRKY domains in perceiving and transducing the immune signals mediated by YrU1. Identification of corresponding effector(s) recognised by YrU1 would be the key to understanding YrU1 molecular mechanism.

## Methods

**Plant materials**. *Triticum urartu* accession PI427328 was collected from Shaqlawa, Iraq. The PI428209, PI428214, PI428294 and G1812 accessions were collected from Mardin, Turkey. The whole-genome shotgun draft sequence of the G1812 accession has been published[24]. The PI428309 accession was collected from El Beqaa, Lebanon. No stripe rust resistance genes were reported in these accessions. The high-density genetic map was generated from the F_2 population derived from a cross between resistant *T. urartu* accession PI428309 and susceptible *T. urartu* accession G1812. The hexaploid winter wheat cultivar mingxian169 (susceptible to stripe rust races CYR17, CYR31, CYR32, CYR33 and V26) was used to maintain the *Pst* race used in the experiments. The susceptible spring common wheat cultivar Bobwhite was used to generate the transgenic plants.

**Stripe rust assays**. *Pst* races CYR17, CYR31, CYR32, CYR33 and V26 were used in the experiments. We inoculated the stripe rust by spraying the mixture of urediospores and talcum powder at a ratio of 1:2 at the two-leaf stage, and talcum powder was used as an indicator of the urediospores on the leaves during the inoculation[9]. Immediately after inoculation, plants were incubated in a greenhouse at 10 °C with 100% relative humidity in the dark for 24 h. The plants were then moved to a greenhouse at 16–18 °C under a 14-h-light/10-h-dark cycle. The disease infection type (IT) was evaluated at 14 dpi. Stripe rust infection types were assessed based on a 0–4 scale[45]. In details, IT 0: immune, no visible uredia and necrosis on leaves; IT: nearly immune, no uredia with hypersensitive flecks on leaves; IT 1: very resistant, few small uredia with distinct necrosis on leaves; IT 2: moderately resistant, few small- to medium sized uredia with dead or chlorosis on leaves; IT 3: moderately susceptible, a lot of medium-sized uredia, no necrosis, but with chlorosis on leaves; IT 4: highest susceptible, a large number of large-sized uredia without necrosis on leaves.

**Microscopy**. To visualise the average infection area, the infected leaves were detached at 2 and 14 dpi and stained with WGA-FITC (L4895-10MG; Sigma) as described previously[46,47], with minor modification. Briefly, the leaves were cut into 2-cm pieces and placed in a 10 ml centrifuge tube with 5 ml of 1 M KOH and 0.05% Silwet L-77. After 12 h, the KOH solution was gently poured off and washed with 10 ml of 50 mM Tris (pH 7.5). This solution was then replaced with another 10 ml of 50 mM Tris (pH 7.5). After 20 min, the Tris solution was removed and replaced with 5 ml of 20 μg ml⁻¹ WGA-FITC. Tissue was stained for 15 min and then washed with 50 mM Tris (pH 7.5). The WGA-FITC-stained tissue was examined under blue light excitation with a Zeiss LSM 880 confocal microscope.

**Development of molecular markers**. The simple sequence repeat (SSR) markers on hexaploid wheat chromosomes (1A to 7A) were used to generate the initial genetic map of *YrU1*. First, the SSR markers were used to screen the polymorphisms between PI428309 and G1812, and then eighty-two polymorphic markers were used to construct linkage map using the susceptible plants of the F_2 population derived from the cross between PI428309 and G1812. The genetic map

was constructed by software JoinMap 4.0. Closely linked flanking markers were developed from the scaffolds of G1812 and designed using SSR Locator software[48] and dCAPS Finder 2.0 (http://helix.wustl.edu/dcaps/dcaps.html). The polymorphic markers were used to screen the susceptible plants of the F_2 population for the recombinants. In total, 3304 susceptible plants were used to map the resistance gene.

**Quantitative real-time RT-PCR**. Total RNA was isolated using TRIzol reagent (Invitrogen). First-strand cDNA from total RNA was synthesized using murine leukemia virus reverse transcriptase (Promega) and quantitative reverse transcription PCR (qRT-PCR) was performed with SYBR green kit (Takara) following the manufacturer's instructions.

**Cloning of the stripe rust resistance gene**. In order to clone the candidate genes in PI428309, we designed primers based on the sequences of *TuG1812G0500003718*, *TuG1812G0500003719* and *TuG1812G0500003720*. We obtained the DNA sequences of two candidate genes in PI428309, which we designated *YrU1* and *CG2* based on subsequent analysis, by PCR amplification using gene specific primers YrU1-C and CG2-C according to the sequences of *TuG1812G0500003718* and *TuG1812G0500003719*, respectively. We obtained the coding sequence of *YrU1* by PCR amplification from the cDNA of PI428309 using the primer YrU1-C, however, we could not obtain the coding sequence of *CG2* as expression of *CG2* was not detected. We also validated the coding sequence of *YrU1* by rapid amplification of 3' and 5' cDNA ends.

**Prediction and annotation of coding sequences of *YrU1***. The predicted open reading frame of the coding sequence based on the sequence of 3' and 5' RACE was performed with the online program (http://www.softberry.com/berry.phtml?topic=fgenesh&group=programs&subgroup=gfind) and translated using the DNAMAN tool (v9.0.1.116). The *YrU1* encoded an ANK-NLR-WRKY protein based on the functional annotation using the SMART program (http://smart.embl-heidelberg.de/), the LRRsearch program (http://lrrsearch.com/) and the CD-search online tool (https://www.ncbi.nlm.nih.gov/Structure/cdd/wrpsb.cgi).

**Wheat transformation**. The 11,858-bp genomic fragment of *YrU1*, including the 2714 bp upstream of the start codon, all the exons and introns and 2086 bp downstream of the stop codon, was cloned into the vector pCAMBIA-1300. The *YrU1* genomic fragment in the pCAMBIA-1300 vector was introduced into the stripe-rust-susceptible wheat cultivar Bobwhite by particle bombardment as described previously[26,49].

**Phylogenetic analyses**. We used the neighbour-joining method implemented in MEGA (v7.0.26) to analyse the relationships between ANK domains from ANK-NLR and non-NLR proteins. The phylogenetic tree analyses were performed as described previously[7], with a little alteration. First, we retrieved all ANK-containing proteins from the *T. urartu* genome and ANK-NLR proteins from grass genomes using the following steps: we used hmmer (v3.1b2, http://hmmer.org/) (UniProt references proteomes) to identify conserved domains in protein sequences from the *T. urartu* genome. We applied default parameters to filter out any unrelated identified domains. ANK domains of the corresponding proteins in the output from hmmer were verified on the CD-search database. We divided the group between ANK-NLRs and non-NLRs based on the presence of NB-ARC domains. The phylogenetic tree analyses were performed in MEGA (v7.0.26). We used the same method to generate a phylogenetic tree of WRKY domains.

**Yeast two-hybrid assays**. Yeast two-hybrid assays were performed in the Matchmaker GAL4 Two-Hybrid System 3 (Clontech). To examine the self-association of the ANK and CC domains of YrU1, the coding sequences of YrU1-ANK and YrU1-CC were cloned into pGADT7 or pGBKT7. Different pairs of constructs were co-transformed into *Saccharomyces cerevisiae* strain AH109.

**Immunoblot analysis**. Different plasmids transformed into *Agrobacterium tumefaciens* GV3101 were suspended in infiltration buffer as previously described[50] to OD_600 = 1.5. Transient expression, total protein extraction and immunoblot analysis were performed as described[51], with minor alterations. Total protein extracts (1 ml) were incubated with 20 μL agarose-conjugated anti-GFP antibody (MBL, D153-8) at 4 °C for 4 h with gentle rotation. After incubation, the agarose beads were washed four times with phosphate-buffered saline (PBS) containing 0.008-0.01% (v/v) Triton X-100 and resuspended in 80 μL PBS; 20 μL 5× SDS/PAGE loading buffer was added and then the mixture was boiled for 5 min. The proteins were detected with anti-MYC (Abmart, M2002), anti-GFP (Abmart, M2004) and anti-HA (Abmart, M2003) immunoblot.

**Luciferase complementation imaging assay**. To examine the self-association of the ANK and CC domains of YrU1, the coding sequences of YrU1-ANK and YrU1-CC were cloned into pCAMBIA1300-nLUC or pCAMBIA1300-cLUC. YrU1-ANK-nLUC/YrU1-ANK-cLUC or YrU1-CC-nLUC/YrU1-CC-cLUC were

cotransformed into *N. benthamiana* by infiltration with Agrobacterium GV3101. After 2 days, 1 mM precooled luciferin was sprayed onto the leaves, and the samples were incubated in the dark for 5–10 min. LUC images were captured using a cooled CCD imaging apparatus[52].

**Oligonucleotide sequences.** Primers used in this study are listed in Supplementary Table 8.

**Reporting summary.** Further information on research design is available in the Nature Research Reporting Summary linked to this article.

## Data availability

Data that supporting the findings of this study are presented in the manuscript and the Supplementary files. Sequence data of *YrU1* gene in PI428309 can be found in the GenBank database under the accession number MT018453. Source data of Figs. 1, 3, 4 and 6, as well as Supplementary Figs. 1-4, 7-9 and 11 are provided as a Source Data file.

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

## Acknowledgements

We thank United States Department of Agriculture (USDA) for providing us with *T. urartu* material. This work was supported by grants from the National Natural Science Foundation of China (31830077) and National Science Fund for Distinguished Young Scholars of China (31525019) to D.T.

## Author contributions

H.W. and D.T. conceived the research; H.W. and S.Z. performed the experiments; H.W. and D.T. analysed the data; Y.L. contributed *Triticum urartu* lines; F.L. performed the rust inoculation; H.W. and D.T. wrote the paper; and all authors contributed to revision of the paper.

## Competing interests

We have filed a patent application based on this work. The patent applicant is Fujian Agriculture and Forestry University and the inventors are Dingzhong Tang, Huan Wang and Shenghao Zou. The application number is 202010072144.3. The patent application has been submitted for consideration, and the cloning and validation of *YrU1* described in this paper are covered in the patent application. Other authors are not listed in the patent application and declare no competing interests.
