## [Peer Review File · Nature Communications]

Reviewers' comments:

Reviewer #1 (Remarks to the Author):

I am indeed impressed by the volume of work presented, and feel it has the potential to be quite novel by revealing the structure of unique gene structure for disease resistance that appears to be unique to wheat. However, after careful review, I conclude that the manuscript is not ready for publication in its current form. The two main claims of the paper, the positional cloning of Yr90 gene and its unique ANK-NLR-WRKY structure, is not fully supported by data presented in the manuscript. Of critical importance is that it is not clear: (i) how the sequence of R-gene was obtained and, (ii) how the novel gene was annotated. Below I also highlight other issues that must be consider prior to resubmission.

1. The candidate genes in PI428309 donor line were obtained through homology-based cloning using the homologous sequence of the susceptible line G1812 There is no explanation in the Results or the M&M sections of the manuscript regarding what is meant by homology-based cloning, or the procedure that was used to clone the resistance gene. It is therefore impossible to assess the reliability of the obtained sequence, which is not presented in the manuscript, its supplementary files and is not deposited to publicly available domains (such as NCBI). Moreover, there is no any information regarding full-length Yr90 genomic sequence and its translated protein sequence in the manuscript, which is supposed to be a base to support statement about cloning of the gene.

2. Identifying a unique gene structure for disease resistance in plants is indeed a novel component of the manuscript. There is reasonable evidence to support that the authors have successfully cloned Yr90 (based on transgenic evidence), but functional annotation is largely lacking. Indeed identifying the unique gene structure only in a specific taxonomic group (here Triticeae) is unusual, which is exciting and can facilitate an understanding of evolutionary processes. However, the annotation of the resistance gene is only based on projection and comparison of annotations from existing wheat reference genomes. No attempt was made to validate these annotations (via RNASeq or other methods) which is absolutely critical when describing a new gene structure. Since the authors showed that ANK domain and NRL-WRKY combination can be found independently in reference genomes of different species and NLR-WRKY combination was proven previously to be functional in Arabidopsis, it is absolutely necessary to provide evidences that ANK-NLR-WRKY structure is indeed real and all domains are necessary for functionality. Unfortunately, this was not done in current research.

3. Following the claim of the presence of ANK-NLR-WRKY only in Triticeae, it would be logical to see as more work as possible done for Triticeae. However, the authors only used *T. dicoccoides* Zavitan and *T. aestivum* Chines Spring reference genomes. It would sensible to conduct a similar survey in *T. durum* Svevo whole genome pseudomolecule assembly and its annotation as well as other wheat genomes available at scaffold level from TGAC (such as Cadenza, Kronos, Paragon, Robigus, Claire). Moreover, Triticeae tribe includes not only Triticum species, but also Aegilops, Hordeum, Secale and others. The search conducted by authors didn't yield genes with ANK-NLR-WRKY structure in other than Triticum species, in particular any ANK-NLR-WRKY structure was detected in Hordeum. Thus, I would suggest to reconsider this claim of the presence of genes with such unique structure only in Triticeae, this rather should be Triticum species.

4. The manuscript presents a tremendous amount of useful information, but would benefit from a through revision to correct technical errors and to clarify results to better support conclusions. For example:

- It seems that Yr90 gene was mapped to 5A chromosome, however it is not mention clearly not in the main text, nor at the Fig. 2 with physical and genetic maps. Moreover, there is no explanation what two lines at Fig. 2 b panel mean. What does the blue line corresponds to? Furthermore, the genetic map shows recombination (0.034 cM) between Yr90 and SCF24 marker (Fig. 2 a), while on physical map (Fig. 2 b) SCF24(1) marker is located inside the Yr90 gene. Are SCF24 and SCF24(1) correspond to the same marker (there is sequence for only one SCF24 marker in Suppl. Table 8)? If

yes, how authors will explain these not matching results from genetic and physical maps?

- It is not clear homologs of which genes were found in R donor: “we obtained two candidate genes in PI428309 allelic to TuG1812G0500003719 and TuG1812G0500003720, which we designated Yr90 and CG2 based on subsequent analysis. There is no allelic gene of TuG1812G0500003720 in PI428309” (lines 138-141). In some parts of the manuscript, TuG1812G0500003718 was actually the Yr90. This presents a great deal of confusion to the reader.

- Authors used infection type (IT) scale from 1 to 4 (according to legend of Fig. 1), however, there is no description what IT1 or IT4 means. Compared to the usual 0-9 IT scale (Line and Qayoum, 1992), it is hard to believe that IT1 in 1-4 scale means already presence of sporulation (which is visible on all phenotype pictures of PI428309). Taking into account the sporulation observed on green leaf area, it is my opinion that PI428309 should be classified as moderate resistant. Moreover, F1 plants showed resistance intermediate between the two parents, supporting incomplete dominance (although they definitely had hypersensitive response). Given this, scoring and phenotype classification in F2 population would be more difficult than presented. Based on parental and F1 plants phenotypic responses, it is unlikely given that all F2 plants showed clear separation into two classes with segregation ratio 3:1. Which ITs corresponded to “resistant” and which to “susceptible”?

- The figures describing the Pst phenotypic responses are unclear and, in my opinion, need to be reobtained to support claimed statements. Specifically: (i) the authors state that “Chinese Spring is susceptible to stripe rust Pst CYR33CS”, while the photo shows an unusually yellow leaf (which could be interpreted by some as a hypersensitive response???) with sporulation (Suppl. Fig. 9); (ii) In Fig. 3 the color of the leaves of BSMV:GFP and BSMV:Yr90 look almost the same, however BSMV:GFP was claimed to express substantial cell death, while BSMV:Yr90 lacked visible cell death (lines 165-166); (iii) It seems that all leaves in Suppl. Fig. 8 have powdery mildew contamination (?) that could influence the results of phenotyping since it is known that plants can become more susceptible to a disease in response to simultaneous infection by multiple-pathogens.

Given my review, I feel the manuscript does not meet the high standards for publication in Nature Communications. However, taking into account that Yr90 may represent a unique R-gene structure of ANK-NLR-WRKY found only in Triticea, the manuscript has potential to influence thinking in the field of plant-pathogen interactions and can be of interest to others in the field. Thus, I strongly encourage the authors to resubmit the manuscript after substantial revision.

Reviewer #2 (Remarks to the Author):

Wang et al describe the discovery and cloning of a novel stripe rust resistance gene, designated Yr90. The unique selling point of this study stems from the fact that Yr90 was found to be an atypical NLR immune receptor with two distinct integrated domains, an Ankaryn domain at the N-terminal and a WRKY domain at the C-terminal. This is the first report of an NLR protein with both N- and C-terminal integrated domains. The discovery and study of NLRs with integrated domains is a hot topic in molecular plant-pathogen interactions and, based on the “integrated decoy” hypothesis is helping to shed light on putative host pathogenicity targets by pathogen effectors. This study by Wang et al. adds to this growing body of knowledge. The study is thorough, very well presented with a logical and easy to follow presentation, both with regards to the text and the figures, and the statements are well supported by the findings. I expect this paper will be very well received and gain attention from the communities studying (i) cereal disease resistance, (ii) the structure, function and evolution of plant disease resistance genes, and (iii) the general molecular plant-pathogen interactions community. In conclusion, I am very supportive of this study. Indeed, I only noticed some very minor issues largely of a typographical nature which will be simple to resolve (as detailed below).

Minor concerns:

Line 38. The authors refer to Yr10 as one of the genes having been cloned and cite: Liu et al., 2014: The stripe rust resistance gene Yr10 encodes an evolutionary-conserved and unique CC-NBS-LRR sequence in wheat. Mol Plant 7:1740-55. However, a follow-up study by Yuan and colleagues (Yuan et al., 2018: Remapping of the stripe rust resistance gene Yr10 in common wheat. TAG 131:1253-1262) subsequently showed the claims of Liu et al to be erroneous. I would therefore suggest that Yr10 be removed from the list of genes mentioned in lines 38-39.

Lines 45 and 59. Yr5, Yr7 and YrSp are not one and the same gene. As shown by Marchal et al. 2018 Nature Plants 4:662-668) Yr5 and Yr7 are distinct, while YrSp is likely a truncated allele of Yr5. Therefore, line 45 should be changed to read: "... and Yr5, Yr7 and YrSp encode..." and line 59 should be changed to read: "the wheat stripe rust resistance proteins Yr5, Yr7 and YrSp have an N-terminal..."

Lines 54 and 55. The abbreviations for NBS and LRR were already introduced in lines 44 and 45, so I see no need to introduce them with longhand again.

Line 61. Direct should be directly.

Line 181. Resistant should be resistance.

Line 189. Should be "The domain structure of the Yr90 protein"

Line 191. This line doesn't sound grammatically correct. Perhaps consider something along the lines of: "... the genomes of all species from the public reference proteome database UniProt."

Line 219. Write out 2 as two. Ditto in Figure 5 legend.

Line 263. Which database? And, please fix the grammar in this sentence.

Line 291. Should be: "... with the effector of the pathogen..."

Line 326. An effector is not born with an R gene, so the terminology 'cognate' is wrong. Please consider replacing 'cognate' with 'corresponding'. For further discussion of this, please see: Oliver (2010): Does a cognate receptor know its own effector? Trends in Plant Sciences 15:539.

Line 367. Should be: "from the T. urartu..."

Figure 2. In the figure legend, please explain the numbers below scaffold 12794. I presume they are the number of recombinants, but please confirm. Also, because the numbers are below the markers, and not between the markers, it is not clear to me how many recombinants there are between the markers.

Figure 6 title. I believe it should be "Self-association" rather than "Self-associate".

Supplementary Tables 2 and 3: It should be "were designed" rather than "that designed".

Supplementary Table 4. Should be "... accessions detected by the Yr90 gene specific marker."

Supplementary Table 5. "Subsp." and "Indica Group" should not be in italics.

Reviewer #3 (Remarks to the Author):

The authors used map-based cloning and isolated a stripe rust resistance gene from the diploid A-genome progenitor *Triticum urartu* of wheat. They named the gene Yr90. However, this does not follow the rules of wheat gene nomenclature. Only the genes that have been introgressed into wheat are given permanent official names of YrXX, which need the approval from the curator of the Catalogue of Gene Symbols for Wheat after consulting with certain group. In this manuscript, the gene was isolated from *T. urartu* and had not been introgressed into wheat, therefore, only temporary name can be used, rather than using the official name for the wheat genes.

Their gene has a unique structure of having an N-terminal ankyrin-repeat and a C-terminal WRKY-domain in addition to the common NLR structure of the resistant gene. The gene was transformed into wheat and was shown to confer resistance. Because current acceptance of GMO or gene-editing is still in question, the authors can consider using conventional cross to transfer this gene into wheat so that it can be a new source of resistance. Overall, the manuscript was well written and the topic is of interest of broader research community. However, there are several issues that need to be addressed before it can be accepted in addition to the ones above.

L41-44: Yr15 is an all-stage resistance gene, rather than an adult plant resistance (APR) gene.

L66-67: There has not been an officially named stripe rust resistance gene introgressed into wheat from *T. urartu*. Therefore, it is not appropriate to state that *T. urartu* is "an important source for stripe rust resistance genes". "a potential source" is better.

In general, the rust infected leaf photos were not in good quality, especially those in Supplementary Fig. 1, 2, 8, 9. Some photos are upper leaves, and some are lower leaves. Leaf segments are too small to see the infection type clearly. In Suppl. Fig. 9, the colours of Chinese Spring leaves and spores are not right.

L105-107: It is better to use "their infection types were similar but slightly higher than those of the PI428309 plants".

L121 and others: It should be "deletion bins", rather than "deletion-line bins".

L143 and others: It is better to use "~1Kb", rather than "~1,000bp".

L174-175: The authors stated that they obtained three independent positive transgenic plants. How many transgenic plants in total they recovered? Because they used the bombardment, most probably the transgenic plants would have more than one copy of the transgene. Have they done any tests on the copy numbers to correlate with the phenotype?

L194-195: These three proteins are actually homologs, rather than alleles.

L229-232: Functional study showed that using an N-terminal tag on the full-length and different domains did not induce cell death. However, it is known that the N-terminus has an important function, having a tag may influence its function, hence, not inducing cell death. Have the authors tried the C-terminal tag? If they have, what are the results?

L329-331, Supplementary Fig. 4 and Supplementary Table 4: Are the *T. urartu* accessions obtained from the USDA National Small Grains Collection, which needs to be acknowledged here?

Supplementary Fig. 7: The SDs are big in almost all the time points, which made the Yr90 expression difference between PI428309 and G1812 not big. The authors need to add the significance level on top of the bars.

Supplementary Table 2: The Deletion bin location for SCF19 was missing, is it "5AL10-0.57-0.78"?

Supplementary Table 4: It is difficult to imagine that so many different *T. urartu* accessions have Yr90 gene marker, if considering different accessions are different. The authors need to look into their collection details (e.g. location) because the *T. urartu* collected from different locations were given different accession numbers, but they might be actually the same. The authors do need to interpret the results in this table sensibly.

Supplementary Table 5: It should be "Species" in the heading of the table.

Response to Reviewer 1:

I am indeed impressed by the volume of work presented, and feel it has the potential to be quite novel by revealing the structure of unique gene structure for disease resistance that appears to be unique to wheat. However, after careful review, I conclude that the manuscript is not ready for publication in its current form. The two main claims of the paper, the positional cloning of Yr90 gene and its unique ANK-NLR-WRKY structure, is not fully supported by data presented in the manuscript. Of critical importance is that it is not clear: (i) how the sequence of R-gene was obtained and, (ii) how the novel gene was annotated. Below I also highlight other issues that must be consider prior to resubmission.

RESPONSE: We thank the Reviewer for the encouraging comments. Please see below for how we obtained the sequence of R-gene, and how the gene was annotated.

1. The candidate genes in PI428309 donor line were obtained through homology-based cloning using the homologous sequence of the susceptible line G1812 There is no explanation in the Results or the M&M sections of the manuscript regarding what is meant by homology-based cloning, or the procedure that was used to clone the resistance gene. It is therefore impossible to assess the reliability of the obtained sequence, which is not presented in the manuscript, its supplementary files and is not deposited to publicly available domains (such as NCBI). Moreover, there is no any information regarding full-length Yr90 genomic sequence and its translated protein sequence in the manuscript, which is supposed to be a base to support statement about cloning of the gene.

RESPONSE: Thanks for the comments. To avoid confusion, we deleted the sentence “Through homology-based cloning” and added the detailed method for cloning of the *Yr90* (*YrU1* in revision) gene in the Methods section in lines 370-380 in the revision. It reads: “In order to clone the candidate genes in PI428309, we designed primers based on the sequences of *TuG1812G0500003718*, *TuG1812G0500003719* and *TuG1812G0500003720*. We obtained the DNA sequences of two candidate genes in PI428309, which we designated *YrU1* and *CG2* based on subsequent analysis, by PCR amplification using gene specific primers YrU1-C and CG2-C according to the sequences of *TuG1812G0500003718* and *TuG1812G0500003719*, respectively. We obtained the coding sequence of *YrU1* by PCR amplification from the cDNA of PI428309 using the primer YrU1-C, however, we could not obtain the coding sequence of *CG2* as expression of *CG2* was not detected. We also validated the coding sequence of *YrU1* by rapid amplification of 3' and 5' cDNA ends.” It is relatively easy to clone the *YrU1* gene, as the sequences of 5' and 3' ends of this R-gene are identical to *TuG1812G0500003718* of the sequenced G1812 genome.

We provided the gene sequence, coding sequence and protein sequence in the File 1 (for review only). Those sequences will be deposit to Genebank (NCBI) when the manuscript is near acceptance.

2. Identifying a unique gene structure for disease resistance in plants is indeed a novel

component of the manuscript. There is reasonable evidence to support that the authors have successfully cloned Yr90 (based on transgenic evidence), but functional annotation is largely lacking. Indeed identifying the unique gene structure only in a specific taxonomic group (here Triticeae) is unusual, which is exciting and can facilitate an understanding of evolutionary processes. However, the annotation of the resistance gene is only based on projection and comparison of annotations from existing wheat reference genomes. No attempt was made to validate these annotations (via RNASeq or other methods) which is absolutely critical when describing a new gene structure. Since the authors showed that ANK domain and NRL-WRKY combination can be found independently in reference genomes of different species and NLR-WRKY combination was proven previously to be functional in Arabidopsis, it is absolutely necessary to provide evidences that ANK-NLR-WRKY structure is indeed real and all domains are necessary for functionality. Unfortunately, this was not done in current research.

RESPONSE: Thanks for the comments and suggestions. We have added the methods for the Prediction and annotation of coding sequences of YrU1 in the Methods section in lines 381-389. It reads, “The predicted open reading frame of the coding sequence based on the sequence of 3’ and 5’ RACE was performed with the online programs (<http://www.softberry.com/berry.phtml?topic=fgenes&group=programs&subgroup=gfind>) and translated using the DNAMAN tool (v9.0.1.116). The *YrU1* encoded an ANK-NLR-WRKY protein based on the functional annotation using the SMART program (<http://smart.embl-heidelberg.de/>), the LRRsearch program (<http://lrrsearch.com/>) and the CD-search online tool (<https://www.ncbi.nlm.nih.gov/Structure/cdd/wrpsb.cgi>).”

3. Following the claim of the presence of ANK-NLR-WRKY only in Triticeae, it would be logical to see as more work as possible done for Triticeae. However, the authors only used *T. dicoccoides* Zavitan and *T. aestivum* Chines Spring reference genomes. It would sensible to conduct a similar survey in *T. durum* Svevo whole genome pseudomolecule assembly and its annotation as well as other wheat genomes available at scaffold level from TGAC (such as Cadenza, Kronos, Paragon, Robigus, Claire). Moreover, Triticeae tribe includes not only *Triticum* species, but also *Aegilops*, *Hordeum*, *Secale* and others. The search conducted by authors didn’t yield genes with ANK-NLR-WRKY structure in other than *Triticum* species, in particular any ANK-NLR-WRKY structure was detected in *Hordeum*. Thus, I would suggest to reconsider this claim of the presence of genes with such unique structure only in Triticeae, this rather should be *Triticum* species.

RESPONSE: Thanks for the suggestions. We performed a BLASTP analysis with the YrU1 protein in *T. durum* Svevo whole genome and did not find any ANK-NLR-WRKY protein. We added this information in lines 188-190 in the revision.

We also performed BLASTN analysis across the genomes of the five additional wheat cultivars (Cadenza, Kronos, Paragon, Robigus, Claire) from the database GrainGenes (<https://wheat.pw.usda.gov/GG3/>) and did not find any

ANK-NLR-WRKY protein, most likely due to the lack of the annotation of the genomic sequences. We thus did not include this information in the revision.

We have changed the “Triticeae” to “Triticum species” in line 190 as suggested.

4. The manuscript presents a tremendous amount of useful information, but would benefit from a through revision to correct technical errors and to clarify results to better support conclusions. For example:

- It seems that Yr90 gene was mapped to 5A chromosome, however it is not mentioned clearly not in the main text, nor at the Fig. 2 with physical and genetic maps. Moreover, there is no explanation what two lines at Fig. 2 b panel mean. What does the blue line correspond to? Furthermore, the genetic map shows recombination (0.034 cM) between Yr90 and SCF24 marker (Fig. 2 a), while on physical map (Fig. 2 b) SCF24(1) marker is located inside the Yr90 gene. Are SCF24 and SCF24(1) correspond to the same marker (there is sequence for only one SCF24 marker in Suppl. Table 8)? If yes, how authors will explain these not matching results from genetic and physical maps?

RESPONSE: Thanks for the suggestions. We have revised the sentence “we ultimately anchored the *Yr* resistance locus to a region of 65 kb in the G1812 genome” to “we ultimately anchored the *Yr* resistance locus to a region of 65 kb in the G1812 chromosome 5AL” in lines 129-130.

We divided the Fig 2b into two panels (NEW Fig 2b and Fig 2c). At NEW Fig. 2b the line represents the scaffold12794 and in NEW Fig. 2c, the line represents the 518845000 bp to 519045000 bp region on the G1812 chromosome arm 5AL. The blue line corresponds to 518910000 bp to 519030000 bp region on the G1812 chromosome arm 5AL, we highlighted it with blue line and diagonals, as no gene was predicted in this region. We have added the explanation of the Fig. 2b and Fig 2c in the legends in NEW Fig. 2.

SCF24 and SCF24 (1) are correspond to the same marker. The distance in the genetic map was based on one recombination between *Yr90* (*YrUI* in NEW manuscript) and SCF24 marker. It is true that this marker is located in the 3 – 4 Kb of the *YrUI*, and there was a recombinant for this marker and the disease phenotype. We sequenced the recombinant for the region between *YrUI* and SCF24 marker, and the result showed that the recombination site occurred in the ~5 Kb of *YrUI*. Consequently, this recombinant (a susceptible individual) has part of *YrUI* from PI428309 and part of *TuG1812G0500003718* from G1812 (a Chimeric gene), which lacks the 1-kb insertion that encodes the WRKY domain. Therefore this recombinant plant did not have a functional *YrUI* gene, and was susceptible to rust.

- It is not clear homologs of which genes were found in R donor: “we obtained two candidate genes in PI428309 allelic to TuG1812G0500003719 and TuG1812G0500003720, which we designated Yr90 and CG2 based on subsequent

analysis. There is no allelic gene of TuG1812G0500003720 in PI428309” (lines 138-141). In some parts of the manuscript, TuG1812G0500003718 was actually the Yr90. This presents a great deal of confusion to the reader.

RESPONSE: Thanks. The alleles of two candidate genes *YrU1* and *CG2* were *TuG1812G0500003718* and *TuG1812G0500003719* in G1812, and the *YrU1* was actually the allele of *TuG1812G0500003718*, we have corrected it in lines 134-136.

- Authors used infection type (IT) scale from 1 to 4 (according to legend of Fig. 1), however, there is no description what IT1 or IT4 means. Compared to the usual 0-9 IT scale (Line and Qayoum, 1992), it is hard to believe that IT1 in 1-4 scale means already presence of sporulation (which is visible on all phenotype pictures of PI428309). Taking into account the sporulation observed on green leaf area, it is my opinion that PI428309 should be classified as moderate resistant. Moreover, F1 plants showed resistance intermediate between the two parents, supporting incomplete dominance (although they definitely had hypersensitive response). Given this, scoring and phenotype classification in F2 population would be more difficult than presented. Based on parental and F1 plants phenotypic responses, it is unlikely given that all F2 plants showed clear separation into two classes with segregation ratio 3:1. Which ITs corresponded to “resistant” and which to “susceptible”?

RESPONSE: We added the infection type (IT) scale of stripe rust in Methods section in lines 344-350. It reads, “Stripe rust infection types were assessed based on a 0– 4 scale (Kang, Z. et al., 2002). In details, IT 0: immune, no visible uredia and necrosis on leaves; IT 1: nearly immune, no uredia with hypersensitive flecks on leaves; IT 2: very resistant, very few small uredia with distinct necrosis on leaves; IT 3: moderately resistant, few small- to medium sized uredia with dead or chlorosis on leaves; IT 4: moderately susceptible, a lot of medium-sized uredia, no necrosis, but with chlorosis on leaves; IT 4: highest susceptible, a large number of large-sized uredia without necrosis on leaves.”

PI428309 was very resistant (IT 1), and F₁ plants were moderately resistant (IT 2). In F₂ plants, IT 1 and 2 were resistant and IT 3 and 4 were susceptible, the ratio of IT 1 and IT 2: IT 3 and IT 4 was 3:1.

- The figures describing the Pst phenotypic responses are unclear and, in my opinion, need to be reobtained to support claimed statements. Specifically: (i) the authors state that “Chinese Spring is susceptible to stripe rust Pst CYR33CS”, while the photo shows an unusually yellow leaf (which could be interpreted by some as a hypersensitive response???) with sporulation (Suppl. Fig. 9); (ii) In Fig. 3 the color of the leaves of BSMV:GFP and BSMV:Yr90 look almost the same, however BSMV:GFP was claimed to express substantial cell death, while BSMV:Yr90 lacked visible cell death (lines 165-166); (iii) It seems that all leaves in Suppl. Fig. 8 have powdery mildew contamination (?) that could influence the results of phenotyping since it is known that plants can become more susceptible to a disease in response to

simultaneous infection by multiple-pathogens.

Given my review, I feel the manuscript does not meet the high standards for publication in Nature Communications. However, taking into account that Yr90 may represent a unique R-gene structure of ANK-NLR-WRKY found only in Triticea, the manuscript has potential to influence thinking in the field of plant-pathogen interactions and can be of interest to others in the field. Thus, I strongly encourage the authors to resubmit the manuscript after substantial revision.

RESPONSE: Thanks for the comments.

(i) We did the experiment again, and replaced the images of the Supplementary Fig. 9 in revision, please see the NEW Supplementary Fig. 9.

(ii) We agree to the Reviewer that the cell death in the leaves was not obvious. We have revised the sentence as it reads “At 14 dpi, the leaf surfaces of BSMV:*YrUI*-treated plants displayed numerous visible uredia, in contrast with the BSMV:*GFP* control plants, which had few uredia.” in lines 160-162.

(iii) The surface of the leaves was talcum powder rather than powdery mildew. We revised the stripe rust assays in the Methods section in lines 338-340. It reads, “We inoculated the stripe rust by spraying the mixture of urediospores and talcum powder at a ratio of 1:2 at the two-leaf stage, and talcum powder was used as an indicator of the urediospores on the leaves during the inoculation (Zhang, C. et al., 2019).” In addition, we repeated the experiment, and replaced the images of the Supplementary Fig. 8 in the revision.

Response to Reviewer 2:

Wang et al describe the discovery and cloning of a novel stripe rust resistance gene, designated Yr90. The unique selling point of this study stems from the fact that Yr90 was found to be an atypical NLR immune receptor with two distinct integrated domains, an Ankaryn domain at the N-terminal and a WRKY domain at the C-terminal. This is the first report of an NLR protein with both N- and C-terminal integrated domains. The discovery and study of NLRs with integrated domains is a hot topic in molecular plant-pathogen interactions and, based on the “integrated decoy” hypothesis is helping to shed light on putative host pathogenicity targets by pathogen effectors. This study by Wang et al. adds to this growing body of knowledge. The study is thorough, very well presented with a logical and easy to follow presentation, both with regards to the text and the figures, and the statements are well supported by the findings. I expect this paper will be very well received and gain attention from the communities studying (i) cereal disease resistance, (ii) the structure, function and evolution of plant disease resistance genes, and (iii) the general molecular plant-pathogen interactions community. In conclusion, I am very supportive of this study. Indeed, I only noticed some very minor issues largely of a typographical nature which will be simple to resolve (as detailed below).

RESPONSE: We thank the Reviewer for the encouraging comments.

Minor concerns:

Line 38. The authors refer to Yr10 as one of the genes having been cloned and cite:

Liu et al., 2014: The stripe rust resistance gene Yr10 encodes an evolutionary-conserved and unique CC-NBS-LRR sequence in wheat. Mol Plant 7:1740-55. However, a follow-up study by Yuan and colleagues (Yuan et al., 2018: Remapping of the stripe rust resistance gene Yr10 in common wheat. TAG 131:1253-1262) subsequently showed the claims of Liu et al to be erroneous. I would therefore suggest that Yr10 be removed from the list of genes mentioned in lines 38-39.

RESPONSE: Thanks for the comments and suggestions. We have removed the *Yr10* from the list of genes in lines 37 and 43 and changed the “Yr7” to “Yr10” in line 49.

Lines 45 and 59. Yr5, Yr7 and YrSp are not one and the same gene. As shown by Marchal et al. 2018 Nature Plants 4:662-668) Yr5 and Yr7 are distinct, while YrSp is likely a truncated allele of Yr5. Therefore, line 45 should be changed to read: “... and Yr5, Yr7 and YrSp encode...” and line 59 should be changed to read: “the wheat stripe rust resistance proteins Yr5, Yr7 and YrSp have an N-terminal...”

RESPONSE: Thanks. We have corrected it as suggested. Please see the lines 37, 44, 57.

Lines 54 and 55. The abbreviations for NBS and LRR were already introduced in lines 44 and 45, so I see no need to introduce them with longhand again.

RESPONSE: Thanks. We have deleted the introduction as suggested in line 53.

Line 61. Direct should be directly.

RESPONSE: Thanks. We corrected it as suggested in line 59.

Line 181. Resistant should be resistance.

RESPONSE: Thanks. We corrected it as suggested in line 176.

Line 189. Should be “The domain structure of the Yr90 protein”

RESPONSE: Thanks. We corrected it as suggested in line 184.

Line 191. This line doesn’t sound grammatically correct. Perhaps consider something along the lines of: “... the genomes of all species from the public reference proteome database UniProt.”

RESPONSE: Thanks. We corrected it as suggested in line 186.

Line 219. Write out 2 as two. Ditto in Figure 5 legend.

RESPONSE: Thanks. We corrected it as suggested in line 216 and Figure 5 legend.

Line 263. Which database? And, please fix the grammar in this sentence.

RESPONSE: Thanks. From the public reference proteome databases UniProt and Ensemble. We have revised the sentence as it reads “we identified only three ANK-NLR-WRKY proteins from the public reference proteome databases UniProt and Ensemble” in lines 259-261.

Line 291. Should be: "... with the effector of the pathogen..."

RESPONSE: Thanks. We corrected it in as suggested lines 287-288.

Line 326. An effector is not born with an R gene, so the terminology 'cognate' is wrong. Please consider replacing 'cognate' with 'corresponding'. For further discussion of this, please see: Oliver (2010): Does a cognate receptor know its own effector? Trends in Plant Sciences 15:539.

RESPONSE: Thanks. We revised the sentence as suggested in line 323.

Line 367. Should be: "from the *T. urartu*..."

RESPONSE: Thanks. We corrected it in line 401 as suggested.

Figure 2. In the figure legend, please explain the numbers below scaffold 12794. I presume they are the number of recombinants, but please confirm. Also, because the numbers are below the markers, and not between the markers, it is not clear to me how many recombinants there are between the markers.

RESPONSE: Thanks. We have added the explanation of the numbers in the legend of Figure 2b. It reads, "The number of recombinants are indicated below the scaffold12794". There are three recombinants between the markers SCF23 and SCF24, one recombinant between the markers SCF24 and *Yr* locus and two recombinants between the markers SCF22 and *Yr* locus.

Figure 6 title. I believe it should be "Self-association" rather than "Self-associate".

RESPONSE: Thanks. We corrected it as suggested.

Supplementary Tables 2 and 3: It should be "were designed" rather than "that designed".

RESPONSE: Thanks. We corrected it as suggested.

Supplementary Table 4. Should be "... accessions detected by the *Yr90* gene specific marker."

RESPONSE: Thanks. We corrected it as suggested.

Supplementary Table 5. "Subsp." and "Indica Group" should not be in italics.

RESPONSE: Thanks. We corrected it as suggested.

Response to Reviewer 3:

The authors used map-based cloning and isolated a stripe rust resistance gene from the diploid A-genome progenitor *Triticum urartu* of wheat. They named the gene *Yr90*. However, this does not follow the rules of wheat gene nomenclature. Only the genes that have been introgressed into wheat are given permanent official names of *YrXX*,

which need the approval from the curator of the Catalogue of Gene Symbols for Wheat after consulting with certain group. In this manuscript, the gene was isolated from *T. urartu* and had not been introgressed into wheat, therefore, only temporary name can be used, rather than using the official name for the wheat genes.

RESPONSE: Thanks for the comments. We contacted Prof. Robert McIntosh, who is in charge of the wheat gene catalog. He read the manuscript, and suggested that the gene name *YrUI* be used for now – when the paper is accepted and the GenBank number is available, he will provide the official name at that time. Here is his response “I read the manuscript and find it interesting as stated by the referees. I believe that if the paper meets acceptance by the journal a unique Yr number can be allocated subject to approval by a panel of YR workers. The gene does not have to be in common wheat, but the germplasm has to be (and is) accessible to others. --- Before naming we will require the GenBank accession number. I suggest that the gene name YrU1 be used for now – when the paper is accepted and the GenBank number is available you can name it with the next available permanent number that I will provide at that time.--- In summary, my response to your request for a gene name is that it can be considered when the paper is near acceptance. I will require more detailed information on the mapping and linkage relationships as well as a GenBank accession number to provide for consideration by a panel of YR workers.”

Their gene has a unique structure of having an N-terminal ankyrin-repeat and a C-terminal WRKY-domain in addition to the common NLR structure of the resistant gene. The gene was transformed into wheat and was shown to confer resistance. Because current acceptance of GMO or gene-editing is still in question, the authors can consider using conventional cross to transfer this gene into wheat so that it can be a new source of resistance. Overall, the manuscript was well written and the topic is of interest of broader research community. However, there are several issues that need to be addressed before it can be accepted in addition to the ones above.

RESPONSE: Thanks for the suggestion. The gene is now on the process of introgression into wheat by crossing.

L41-44: Yr15 is an all-stage resistance gene, rather than an adult plant resistance (APR) gene.

RESPONSE: Thanks. We revised the sentence as it reads “These genes encode different protein families and confer a relatively broad spectrum of resistance” in lines 41-42.

L66-67: There has not been an officially named stripe rust resistance gene introgressed into wheat from *T. urartu*. Therefore, it is not appropriate to state that *T. urartu* is “an important source for stripe rust resistance genes”. “a potential source” is better.

RESPONSE: Thanks. We corrected it as suggested in line 65.

In general, the rust infected leaf photos were not in good quality, especially those in

Supplementary Fig. 1, 2, 8, 9. Some photos are upper leaves, and some are lower leaves. Leaf segments are too small to see the infection type clearly. In Suppl. Fig. 9, the colours of Chinese Spring leaves and spores are not right.

RESPONSE: Thanks. We repeated the phenotypic identification experiments and photographed the infected leaves. The results were shown in New Supplementary Fig. 1, 2, 8, 9.

L105-107: It is better to use “their infection types were similar but slightly higher than those of the PI428309 plants”.

RESPONSE: Thanks. We have modified the sentence to “their infection types were similar but slightly higher than those of the PI428309 plants” as suggested in lines 103-104.

L121 and others: It should be “deletion bins”, rather than “deletion-line bins”.

RESPONSE: Thanks. We corrected it as suggested in line 118.

L143 and others: It is better to use “~1Kb”, rather than “~1,000bp”.

RESPONSE: Thanks. We corrected it as suggested in lines 139, 142 and 144.

L174-175: The authors stated that they obtained three independent positive transgenic plants. How many transgenic plants in total they recovered? Because they used the bombardment, most probably the transgenic plants would have more than one copy of the transgene. Have they done any tests on the copy numbers to correlate with the phenotype?

RESPONSE: Thanks. We obtained six transgenic plants in total. We provided this information in the Results section in lines 169-172. We examined the relative transcript levels of *YrU1* in transgenic plants by qRT-PCR, but we did not examine the copy numbers in the transgenic plants.

L194-195: These three proteins are actually homologs, rather than alleles.

RESPONSE: Thanks. We have corrected “allelic” to “homologous” as suggested in line 192.

L229-232: Functional study showed that using an N-terminal tag on the full-length and different domains did not induce cell death. However, it is known that the N-terminus has an important function, having a tag may influence its function, hence, not inducing cell death. Have the authors tried the C-terminal tag? If they have, what are the results?

RESPONSE: Thanks for the comments and suggestions. We also transiently overexpressed *YrU1* full-length and different domains with C-terminal HA tag in *Nicotiana benthamiana* and found that none of these domains induced cell death. Please see the NEW Supplementary Fig. 11d, e in the revision. Accordingly, we also modified the sentence to “we transiently expressed *YrU1* with an N-terminal GFP tag or C-terminal HA tag under the control of the 35S promoter in *Nicotiana benthamiana*”

in lines 227-229.

L329-331, Supplementary Fig. 4 and Supplementary Table 4: Are the *T. urartu* accessions obtained from the USDA National Small Grains Collection, which needs to be acknowledged here?

RESPONSE: Thanks. The *T. urartu* accessions were obtained from the U.S. National Plant Germplasm System (NPGS), which is managed by the Agricultural Research Service (ARS), the in-house research agency of the United States Department of Agriculture (USDA). We acknowledged United States Department of Agriculture (USDA) in the Acknowledgements.

Supplementary Fig. 7: The SDs are big in almost all the time points, which made the Yr90 expression difference between PI428309 and G1812 not big. The authors need to add the significance level on top of the bars.

RESPONSE: Thanks. There were statistically significant differences in the expression of *YrU1* between PI428309 and G1812 in 2 dpi and 3 dpi ($P < 0.05$), and the statistical significance was indicated in the NEW Supplementary Fig. 7.

Supplementary Table 2: The Deletion bin location for SCF19 was missing, is it “5AL10-0.57-0.78”?

RESPONSE: Thanks. The SCF19 was located in “5AL10-0.57-0.78”, we have added it in the NEW Supplementary Table 2.

Supplementary Table 4: It is difficult to imagine that so many different *T. urartu* accessions have Yr90 gene marker, if considering different accessions are different. The authors need to look into their collection details (e.g. location) because the *T. urartu* collected from different locations were given different accession numbers, but they might be actually the same. The authors do need to interpret the results in this table sensibly.

RESPONSE: Thanks. We retrieved the provenance of these *T. urartu* material in Genesys (<https://www.genesys-pgr.org/zh/>). The *T. urartu* accessions that have YrU1 gene were collected from Armenia, Iran, Jordan, Lebanon, Syria and Turkey and most of them were collected from Lebanon and Syria. We added the provenance of these *T. urartu* material in the NEW Supplementary Table 4.

Supplementary Table 5: It should be “Species” in the heading of the table.

RESPONSE: Thanks. We corrected it as suggested.

REVIEWERS' COMMENTS:

Reviewer #1 (Remarks to the Author):

I have gone over the replies and changes made in response to my comments on the previous submission and I am happy with the way the authors have addressed all questions. I have no further comments or concerns to raise in relation to this much improved manuscript.

Reviewer #3 (Remarks to the Author):

The authors addressed most of the questions raised by the reviewers and revised the manuscript according to the suggestions. English has been greatly improved also. The topic provides new insights on the resistance gene structure, possibly how plant and pathogen interact, and is of interest of broader research community. I recommend that the revised manuscript can be published.

It was good that the authors clarified the rules on naming wheat gene with Prof. Robert McIntosh, who is one of the major curators for the wheat gene catalogue, and the gene was given a temporary name of YrU1. I agree with Prof. McIntosh that the gene can be named before introgressed into wheat because the A-genome chromosomes in *Triticum urartu* can recombine with those in wheat due to the homology.

The quality of the photographs has been improved and now they are much clearer.

Supplementary Table 4: It is better to change "Provenance of material" to "Country of collection".

Response to Reviewer #1:

I have gone over the replies and changes made in response to my comments on the previous submission and I am happy with the way the authors have addressed all questions. I have no further comments or concerns to raise in relation to this much improved manuscript.

RESPONSE: We thank the Reviewer for the encouraging comments.

Response to Reviewer #3:

The authors addressed most of the questions raised by the reviewers and revised the manuscript according to the suggestions. English has been greatly improved also. The topic provides new insights on the resistance gene structure, possibly how plant and pathogen interact, and is of interest of broader research community. I recommend that the revised manuscript can be published.

RESPONSE: We thank the Reviewer for the encouraging comments.

It was good that the authors clarified the rules on naming wheat gene with Prof. Robert McIntosh, who is one of the major curators for the wheat gene catalogue, and the gene was given a temporary name of YrU1. I agree with Prof. McIntosh that the gene can be named before introgressed into wheat because the A-genome chromosomes in *Triticum urartu* can recombine with those in wheat due to the homology.

The quality of the photographs has been improved and now they are much clearer.

Supplementary Table 4: It is better to change “Provenance of material” to “Country of collection”.

RESPONSE: Thanks. We revised it as suggested.